# ReVideo: Remake a Video with Motion and Content Control

**Chong Mou**[1,2,4]**, Mingdeng Cao**[3,4]**, Xintao Wang**[3*]**, Zhaoyang Zhang**[3]**,**
**Ying Shan**[3]**, Jian Zhang**[1,2,4*]

[1]School of Electronic and Computer Engineering, Peking University
[2]Peking University Shenzhen Graduate School-Rabbitpre AIGC Joint Research Laboratory
[3]ARC Lab, Tencent PCG    [4] University of Tokyo
[4]Guangdong Provincial Key Laboratory of Ultra High Definition Immersive Media Technology
`https://mc-e.github.io/project/ReVideo/`

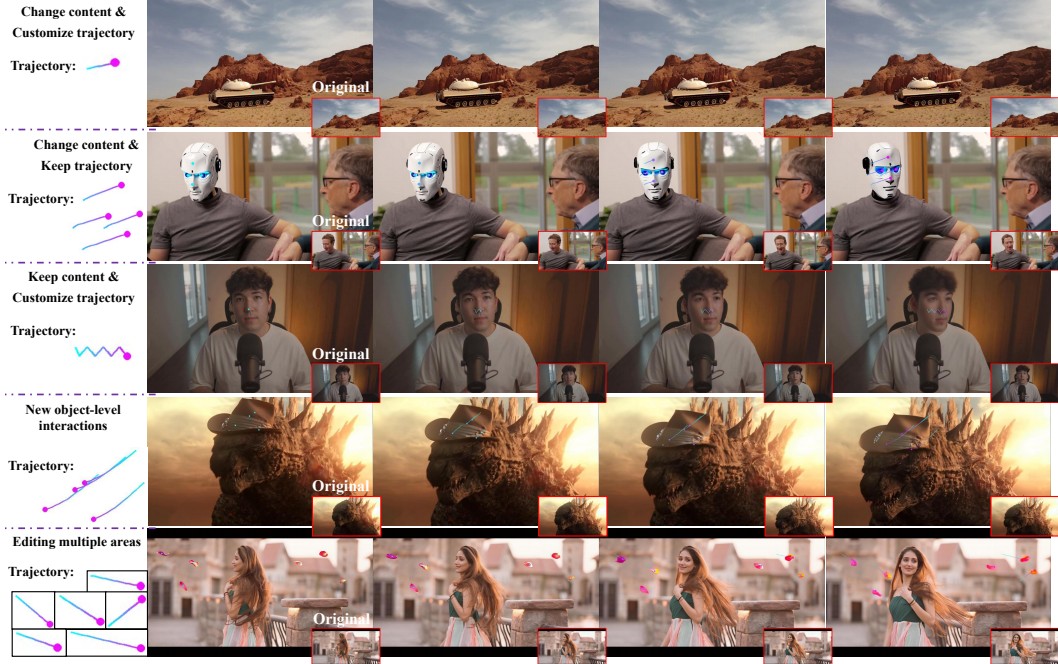

Figure 1: The capability of our method to locally modify video content and motion. This ability can also be easily extended to multi-area editing. The motion control is labeled in colorful lines in videos.

## Abstract

Despite significant advancements in video generation and editing using diffusion models, achieving accurate and localized video editing remains a substantial challenge. Additionally, most existing video editing methods primarily focus on altering visual content, with limited research dedicated to motion editing. In this paper, we present a novel attempt to **Re**make a **Video** (**ReVideo**) which stands out from existing methods by allowing precise video editing in specific areas through the specification of both content and motion. Content editing is facilitated by modify-

---

*Corresponding author

This work was also supported by the National Natural Science Foundation of China (Grant No. 62372016)and the Guangdong Provincial Key Laboratory of Ultra High Definition Immersive Media Technology(Grant No.2024B1212010006).

38th Conference on Neural Information Processing Systems (NeurIPS 2024).

ing the first frame, while the trajectory-based motion control offers an intuitive user interaction experience. ReVideo addresses a new task involving the coupling and training imbalance between content and motion control. To tackle this, we develop a three-stage training strategy that progressively decouples these two aspects from coarse to fine. Furthermore, we propose a spatiotemporal adaptive fusion module to integrate content and motion control across various sampling steps and spatial locations. Extensive experiments demonstrate that our ReVideo has promising performance on several accurate video editing applications, *i.e.*, (1) locally changing video content while keeping the motion constant, (2) keeping content unchanged and customizing new motion trajectories, (3) modifying both content and motion trajectories. Our method can also seamlessly extend these applications to multi-area editing without specific training, demonstrating its flexibility and robustness.

# 1 Introduction

Thanks to the large-scale training data and huge computing power, there have been significant advancements in diffusion-based [19] image and video generation. For personalization purposes, many works add control signals to the generation process, such as text-guided image [39, 40, 38] and video [18, 17, 13, 45] generation, as well as image-guided video generation [4, 54]. Based on these base models, extensive works explore how to transfer their generation capabilities to video editing. Early works based on text-to-image diffusion models implement video editing through zero-shot strategies (*e.g.*, Fate-Zero [36], Flatten [11]) or one-shot tuning (*e.g.*, Tune-A-Video [51]). However, these methods are limited by excessive manual design and a lack of video generation priors. Moreover, text prompt only provide coarse condition, limiting the editing accuracy. Compared to text, more recent methods adopt image conditions which can provide more accurate editing guidance. For instance, VideoComposer [47] generates style-transformed videos by providing spatial attributes (*e.g.*, edge, depth) of the target video and a style reference. DreamVideo [49] and Make-a-protagonist [62] can modify a specific object in the video by providing a reference object. However, these methods still struggle with local editing and introducing new elements, such as adding new objects to a video. Recent work EVE [42] proposes a diffusion distillation strategy to achieve video editing while keeping unedited content unchanged. Nevertheless, the editing region and target are controlled by text, which is challenging in complex scenarios. AnyV2V [26] edit a video by modifying the first frame, enabling accurate customization of local content. Pika [1] can regenerate a specific area in the video by selecting an editing region. Although these methods improve the performance of local video editing, they only focus on visual content editing and cannot customize the motion of new content.

Motion is another crucial aspect of video, yet research on video motion editing remains limited. While some methods explore motion-guided video generation using trajectory-based motion guidance (*e.g.*, DragNUWA [57], DragAnything [52], MotionCtrl [48], VideoSwap [16]) and box-based motion guidance (*e.g.*, Boximator [44], Peekaboo [20]), they do not support motion editing. Additionally, other works [56, 30, 61] can transfer motion from one video to another but cannot modify it as well.

In this paper, our goal is to accurately edit content and motion in specific areas of a video. We create an easy-to-interact pipeline by setting the content editing as modifying the first frame, with trajectory lines [57] as the motion control signal. Other unedited content in all frames should be maintained in editing results and merged with the editing effect. However, we find that fusing unedited content with motion-customized new content is challenging, mainly for two reasons: **(1)** Training imbalance: Unedited content is dense and easier to learn, while motion trajectories are sparse and abstract, making them harder to learn. **(2)** Condition coupling: Unedited content provides both visual and inter-frame motion information, leading the model to rely on it for motion estimation, thereby ignoring the hard-to-learn trajectory lines.

To address these challenges, we design a three-stage training strategy to harmonize unedited content and motion-customized new content, enabling harmonious control of different conditions. Besides, we design a spatiotemporal adaptive fusion module to fuse these two conditions at different diffusion sampling steps and spatial locations. Furthermore, our method can compactly inject motion and content conditions into the diffusion video generation through a single control module. With these techniques, users can conveniently edit specific regions in the video by modifying the first frame and drawing trajectory lines. Notably, ReVideo is not limited to single-region editing and can customize multiple areas in parallel.

In summary, this work makes the following contributions:

- To the best of our knowledge, this is the first attempt to explore local editing of both content and motion in videos. Our method can also be easily extended to multi-area video editing.

- We propose a three-stage training strategy and a spatiotemporal adaptive fusion module to address the coupling of content and motion control in video editing, enabling compact control through a single module.

- Extensive experiments demonstrate that ReVideo performs well in several precise video editing applications, including changing content in a specific region while keeping motion constant, maintaining content while customizing new motion trajectories, and modifying both content and motion trajectories. Some examples are presented in Fig. 1.

## 2 Related Works

### 2.1 Controllable Image and Video Generation

Recent advancements in diffusion models [19, 12] drive the rapid development of image and video generation. In the community of image generation, some notable works, such as Stable Diffusion [39], Imagen [40], and DALL-E2 [38], utilize text as the generation condition. To achieve accurate generation control, some methods, *e.g.*, ControlNet [58] and T2I-Adapter [34], propose adding control modules on pre-trained diffusion models. Similarly, initial efforts in controllable video generation concentrate on the text condition, such as Video LDM [5], Imagen Video [18], VideoCrafter [8], and AnimateDiff [17]. Recognizing the limitations of text prompts in capturing complex scenarios, some recent works [4, 54, 59, 13] leverage image conditions for a more direct approach. External control modules on pre-trained foundation models are also popular in controllable video generation. Such as video ControlNet [9, 60] extends the ControlNet [58] in image generation to video generation conditioned on a sequence of control signals, like edge maps and depth maps. In addition to spatial structure control, precise temporal motion control is also important in controllable video generation. This process is similar to the drag-based image editing [35, 32, 33]. Several recent works study this topic, such as video generation with trajectory-based motion guidance (*e.g.*, DragNUWA [57], MotionCtrl [48], Motion-I2V [41], DragAnything [52]) and generation with box-based motion guidance (*e.g.*, TrailBlazer [29], Boximator [44], [20]). These methods perform the control by training extra motion controllers on pre-trained video diffusion models.

### 2.2 Diffusion-based Video Editing

Due to the lack of training data, the common approach in video editing is via training-free strategies [7, 15, 21, 25, 46, 36] or one-shot tuning [51, 3, 24]. For instance, the prior work Tune-A-Video [51] overfits some diffusion model parameters to a specific video. Then, it uses the overfitting parameters to produce the editing result conditioned on the target prompt. To enable a cohesive global appearance among edited frames, many methods extend the attention module of Stable Diffusion [39] to encompass multiple frames and conduct cross-frame attention. For instance, Pix2Video [7] edits the first frame and performs cross-frame attention of each frame on the first frame to preserve appearance consistency. TokenFlow [15] and Fairy [50] jointly edit a few key frames at each denoising step and propagate them throughout the video based on the nearest-neighbor field extracted from the original video. Inspired by the initial zero-shot image editing method SDEdit [31], the recent video foundation model SORA [6] achieves video editing by adding noise to the input video and then denoising it under the target description. Although these methods can preserve the general structure of original videos, the information loss and the lack of consistency constraints on the original video make them unfit for precise video editing but suitable for global editing like style transfer.

Another strategy is to train a control module to guide the generation with some characters that should persist in the editing result, such as depth [14, 27, 53], sketch [47], and optical flow [55]. However, existing methods primarily focus on preserving spatial structure and are unsuitable for precise video editing. In the community of precise video editing, some works, such as InsV2V [10] and the recent EVE [42], edit the video by providing editing instructions. However, the text-based editing instruction struggles to locate a target region in some complex scenarios. AnyV2V [26] can edit a video by editing the first frame. Pika [1] is designed to regenerate a selected area in a video by text guidance.

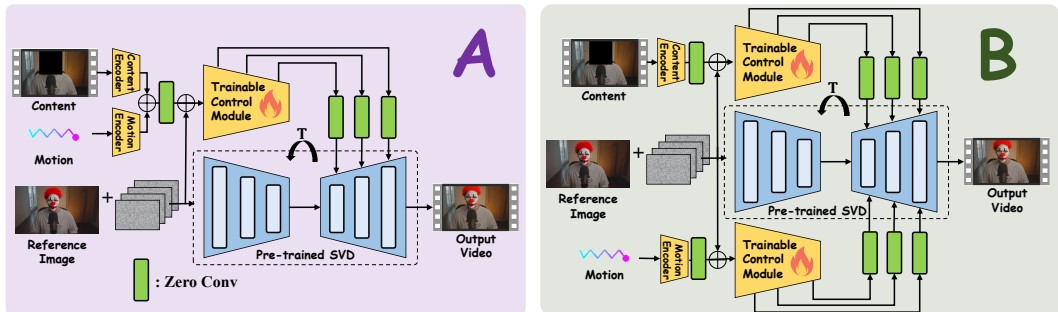

Figure 2: Two potential structures to inject motion and content control. Structure A is a compact and efficient mode that integrates motion and content control via a single module. Structure B features independent control, structurally decoupling motion and content conditions, causing higher complexity.

Unlike these works, we aim to achieve accurate customization in local areas of a video. The editing target includes locally modifying content and motion and keeping the unedited content unchanged.

## 3 Method

### 3.1 Preliminaries

**Stable Video Diffusion** (SVD) [4] is a high-quality and commonly used image-to-video generation model. To utilize the priors of high-quality video generation, we employ SVD as the base model and add control modules to achieve our editing target. Given a reference image $c_I$, SVD will generate a video frame sequence $\mathbf{x} = \{\mathbf{x}^0, \mathbf{x}^1, ..., \mathbf{x}^{N-1}\}$ of length $N$, starting with $c_I$. The sampling of SVD is conducted on a latent denoising diffusion process [39]. At each denoising step, a conditional 3D UNet $\Phi_\theta$ is used to iteratively denoise this sequence:

$$\hat{\mathbf{z}}_0 = \Phi_\theta(\mathbf{z}_t, t, \mathbf{c}_I), \tag{1}$$

where $\mathbf{z}_t$ is the latent representation of $\mathbf{x}_t$. $\hat{\mathbf{z}}_0$ is the predication of $\mathbf{z}_0$. There are two condition paths for the reference image $\mathbf{c}_I$: (1) It is embedded into tokens by CLIP [37] image encoder and injected into the diffusion model vis cross-attention [39]; (2) It is encoded into a latent representation by the VAE encoder, and concatenated with the latent of each frame in channel dimension. SVD follows the EDM-preconditioning framework [23], which parameterizes the learnable denoiser $\Phi_\theta$ as:

$$\Phi_\theta(\mathbf{z}_t, t, \mathbf{c}_I; \sigma) = c_{skip}(\sigma)\mathbf{z}_t + c_{out}(\sigma)F_\theta(c_{in}(\sigma)\mathbf{z}_t, t, \mathbf{c}_I; c_{noise}(\sigma)), \tag{2}$$

where $\sigma$ is the noise level, and $F_\theta$ is the network to be trained. $c_{skip}$, $c_{out}$, $c_{in}$, and $c_{noise}$ are preconditioning hyper-parameters. $\Phi_\theta$ is trained via denoising score matching (DSM):

$$\mathbb{E}_{\mathbf{z}_0, t, \mathbf{n} \sim \mathcal{N}(0, \sigma^2)} \left[ \lambda_\sigma ||\Phi_\theta(\mathbf{z}_0 + \mathbf{n}, t, \mathbf{c}_I) - \mathbf{z}_0||_2^2 \right]. \tag{3}$$

### 3.2 Task Formulation and Some Insights

**Task formulation**. The purpose of this paper is to locally edit a video, including visual information and motion information. In addition, the unedited content in the video should remain unchanged. Therefore, our conditional video generation involves three control signals: (1) the edited content, (2) the content of the unedited area, and (3) the motion condition in the edited area. We implement content editing by modifying the first frame of the video and then broadcasting it to subsequent video frames. Here, we denote the edited first frame as $\mathbf{c}_{ref} \in \mathbb{R}^{3 \times W \times H}$. For the motion condition, we use interaction-friendly trajectory lines [57, 52] as the control signal. Specifically, the motion condition also contains N maps for a N-frame video. Each map consists of 2 channels, indicating the movement of the tracked points in the horizontal and vertical directions relative to the previous frame. The motion condition in this paper is represented as $\mathbf{c}_{mot} \in \mathbb{R}^{N \times 2 \times W \times H}$. The unedited content $\mathbf{c}_{con}$ can be conveniently provided by the masked video, i.e., $\mathbf{c}_{con} = \mathbf{V} \cdot \mathbf{M}$, where $\mathbf{V} \in \mathbb{R}^{N \times 3 \times W \times H}$ and $\mathbf{M} \in \mathbb{R}^{1 \times 1 \times W \times H}$ refer to the original video and the editing region mask, respectively.

Since we adopt SVD as the pre-trained base model, its image-to-video capability can naturally serve as the import port for the edited first frame. For unedited content and customized motion trajectories, we train additional control modules to import them into the generation process.

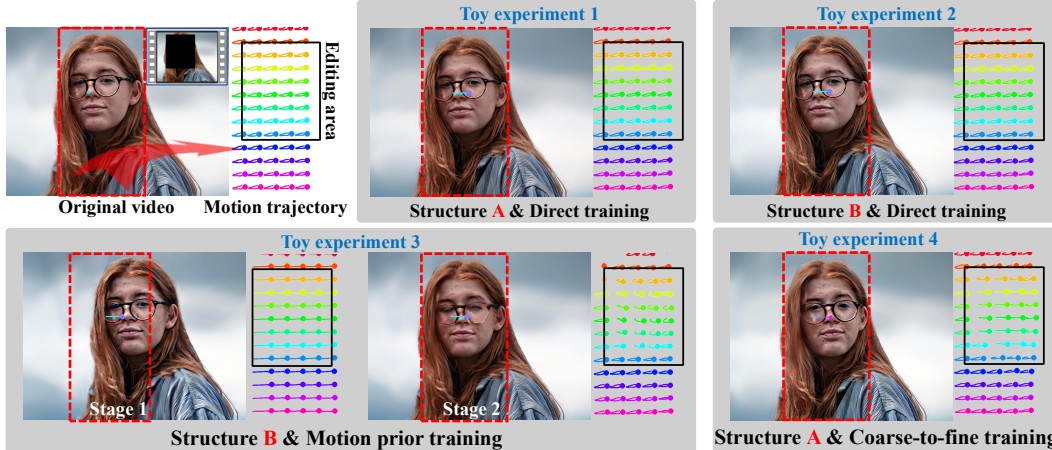

Figure 3: The motion control capability of two structures in Fig. 2 with different training strategies. We visualize trajectory lines in a specific area (red box) and label the editing area with a black box. Toy experiments present the coupling issue of customized motion and unedited content.

**Trajectory sampling**. During training, it is essential to extract trajectories from videos to provide motion condition $\mathbf{c}_{mot}$. At the beginning of trajectory sampling, we use a grid [57] to sparsify dense sampling points, obtaining $N_{init}$ initial points. Among these points, those with larger motions are beneficial to train trajectory control. To filter out these points, we first apply motion tracking on each point to obtain their path lengths, *i.e.*, $\{l_0, l_1, ..., l_{N_{init}-1}\}$. We use the mean of these lengths as the threshold $l_{Th}$ to extract points whose motion length is greater than $l_{Th}$. Then, we use the normalized lengths of these points as sampling probabilities to sample $N$ points randomly. Because the high sparsity is not conducive for the model to learn from these trajectories, we apply a Gaussian filter [57] to obtain the smooth trajectory map $\mathbf{c}_{mot}$. More details are presented in **Appendix**.

**Insights**. A naive implementation of our editing target is directly training an extra control module, like ControlNet [58], to inject motion and content conditions into the diffusion generation process. We present this design in structure **A** of Fig. 2. Specifically, at the input, a content encoder $E_c$ and a motion encoder $E_m$ embed the content condition $\mathbf{c}_{con}$ of the unedited area and motion condition $\mathbf{c}_{mot}$ of the editing area. These two embeddings are merged by direct summing to obtain the fused condition feature $\mathbf{f}_c$. Then, a copy of the UNet encoder extracts multiscale intermediate features from $\mathbf{f}_c$, which are added to the corresponding layers in the diffusion model. This process is formulated as:

$$\mathbf{y}_c = \mathcal{F}(\mathbf{z}_t, t, \mathbf{c}_{ref}; \Theta) + \mathcal{Z}(\mathcal{F}(\mathbf{z}_t + \mathcal{Z}(\mathbf{f}_c), t, \mathbf{c}_{ref}; \Theta_c)), \tag{4}$$

where $\mathbf{y}_c$ is the new diffusion features. $\mathcal{Z}$ is the function of zero-conv [58]. $\Theta$ and $\Theta_c$ are the parameters of the SVD model and extra control module. We conduct several toy experiments based on this idea, as illustrated in Fig.3. The input video contains a woman initially moving to the left, followed by a shift to the right. The editing target is to alter the facial motion towards the right while keeping the other content unchanged. In the toy experiment 1, we fix SVD and train the control module with Eq. 3. The result shows that the content condition precisely controls the unedited area of the generated video. But the motion condition has no control effect, and the trajectory lines in the editing area (labeled with a black box) are consistent with the unedited area. A possible reason is that a single control branch has difficulty handling two control conditions simultaneously. To verify this hypothesis, we train structure **B** in Fig. 2 to handle these two conditions separately. The toy experiment 2 in Fig. 3 shows that the motion control is still ineffective, suggesting that the problem is more attributed to the control training rather than the network structure. To enhance the motion control training, we split the training of structure **B** into two stages. In the first stage, we only train the motion control module to endow it with motion control prior. In the second stage, we train the motion control and content control together. The result in toy experiment 3 shows that although the motion prior training produces good motion control capability, the control accuracy is weakened and affected by the unedited content after introducing the content control. After these toy experiments, we have the following insights:

$\diamond$ The condition of unedited content not only contains visual information but also has rich inter-frame motion information. As a more easily learned condition, the diffusion model tends to predict the motion of the editing area through unedited content, ignoring the sparse motion trajectory control.

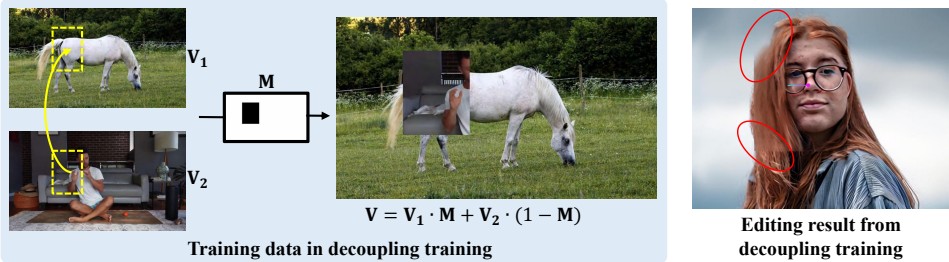

Figure 4: The data construction strategy for decoupling training and editing results from this stage.

◇ The coupling between motion-customized new content and unedited content is strong, making it difficult to overcome even using the motion prior and separate control branches.

◇ Motion prior training is helpful in decoupling motion-customized content and unedited content.

### 3.3 Coarse-to-fine Training Strategy

To rectify the ignoring of the motion control, we design a coarse-to-fine training strategy. In addition, structure B in Fig. 2 has a high computational cost, and we hope to joint control the unedited content and motion-customized new content on the concise structure A.

**Motion prior training**. As discussed above, motion trajectory is a sparse and difficult-to-learn control signal. Toy experiment 3 in Fig. 3 shows that the motion prior training can alleviate the coupling between motion-customized content and unedited content. Hence, in the first stage, we only train the motion trajectory control, allowing the control module to have good motion control prior.

**Decoupling training**. Based on the control module from the first stage, the training in the second stage aims to add content control of unedited areas. Toy experiment 3 in Fig. 3 shows that even with good motion control priors, the precision of motion control still degrades after introducing unedited content condition. Therefore, we design a training strategy to decouple motion and content control in this stage. Specifically, we set the editing part and the unedited part in a training sample $\mathbf{V}$ to be two different videos, *i.e.*, $\mathbf{V}_1$ and $\mathbf{V}_2$. As shown in Fig. 4, $\mathbf{V}_1$ and $\mathbf{V}_2$ are combined through the editing mask $\mathbf{M}$, *i.e.*, $\mathbf{V} = \mathbf{V}_1 \cdot \mathbf{M} + \mathbf{V}_2 \cdot (1 - \mathbf{M})$. Since the editing region and the unedited region come from two different videos, the motion information of the editing region cannot be predicted through the unedited content. Therefore, it can decouple content control and motion control during training.

**Deblocking training**. As shown in the right part of Fig. 4, although the decoupling training achieves joint control of customized motion and unedited content with high accuracy, it breaks the consistency between the edited and unedited regions, producing block artifacts in the boundary. To rectify this issue, we design the third training stage to remove block artifacts. The training in this stage is initialized with the model from the second stage and trained on normal video data. To preserve the decoupled motion and content control prior from the second stage, we only fine-tune the key embedding $\mathbf{W}_k$ and value embedding $\mathbf{W}_v$ in temporal self-attention layers of the control module and SVD model. The toy experiment 4 in Fig. 3 shows that after the training of this stage, the model removes the block artifacts and retains joint control of unedited content and motion customization.

### 3.4 Spatiotemporal Adaptive Fusion Module

Although the coarse-to-fine training strategy achieves decoupling of content control and motion control, we observe considerable failure cases in some complex motion trajectories. To further distinguish the control roles of unedited content and motion trajectories in the generation, we design a spatiotemporal adaptive fusion module (SAFM) as shown in Fig. 5. Specifically, SAFM predict a weight map $\mathbf{\Gamma}$ through the editing mask $\mathbf{M}$ to fuse motion and content control instead of direct summing. Moreover, because diffusion generation is a multi-step iterative process, the fusion of control conditions between time steps should have adaptive adjustment. Therefore, we concatenate timestep $t$ and $\mathbf{M}$ in the channel dimension to form a spatiotemporal condition to guide the $\mathbf{\Gamma}$ prediction. Mathematically, the fusion of motion and content conditions is formulated as follows:

$$\mathbf{f}_c = E_c(\mathbf{c}_{con}) \cdot \mathbf{\Gamma} + E_m(\mathbf{c}_{mot}) \cdot (1 - \mathbf{\Gamma}), \quad \mathbf{\Gamma} = \mathcal{H}(\mathbf{M}, t), \tag{5}$$

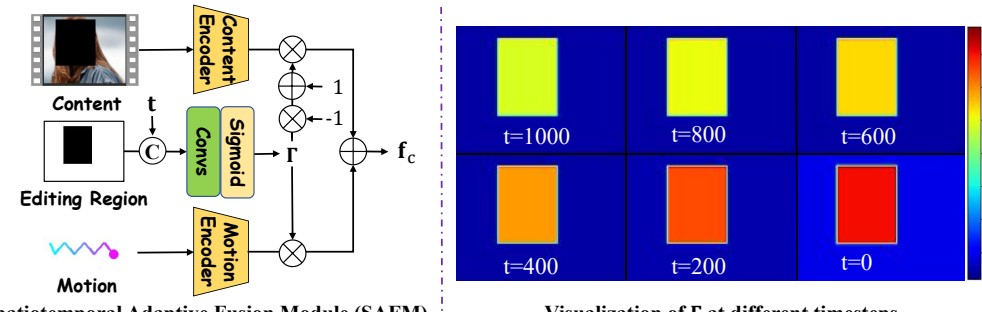

Figure 5: The architecture of our proposed spatiotemporal adaptive fusion module (left), and the visualization of fusion weight $\mathbf{\Gamma}$ at different timesteps (right).

Table 1: Quantitative comparison between our ReVideo and other related works. We employ automatic metrics (*i.e.*, CLIP [37] score, PSNR) and human evaluation to evaluate the performance.

| Method | Automatic Metrics | | | Human Evaluation | | Time $\downarrow$ |
|---|---|---|---|---|---|---|
| | PSNR $\uparrow$ | Text Alignment $\uparrow$ | Consistency $\uparrow$ | Overall $\uparrow$ | Editing Target $\uparrow$ | |
| InsV2V [10] | 29.77 | 0.2022 | 0.9808 | 10.2% | 5.1% | 132s |
| AnyV2V [26] | 29.80 | 0.2143 | 0.9836 | 2.8% | 4.0% | 380s |
| Pika [1] | **33.07** | 0.2184 | **0.9956** | 27.9% | 23.9% | - |
| ReVideo | 32.85 | **0.2304** | 0.9864 | **59.1%** | **67.0%** | **26**s |

where $\mathcal{H}$ is the function of spatiotemporal embedding. $\mathcal{H}$ needs to be jointly trained with $\mathbf{W}_k$ and $\mathbf{W}_v$ in the deblocking training stage. We visualize $\mathbf{\Gamma}$ at different time steps in the right part of Fig. 5. It can be seen that $\mathbf{\Gamma}$ learns the spatial characteristics of the editing area. It assigns a higher weight to the motion condition in the editing area and a higher weight to the content condition in the unedited area. In addition, $\mathbf{\Gamma}$ learns to distinguish different sampling steps $t$ and linearly adjusts with $t$.

## 4 Experiments

### 4.1 Implementation Details

In this work, we choose Stable Video Diffusion (SVD) as the base model. Our three training stages are completed on the WebVid [2] dataset, which contains 10 million text-video pairs. These three stages are optimized for $40K$, $30K$, and $20K$ iterations, respectively, with Adam [28] optimizer on 4 NVIDIA A100 GPUs. The batch size for each GPU is set as 4, with the resolution being $512 \times 320$. It takes about 6 days to complete all training stages. During the training process, we use CoTracker [22] to extract motion trajectories. In the first training stage, trajectory sampling is performed throughout the video. In the second and third training stages, a rectangular editing area is randomly selected in the video with the minimum size being $64 \times 64$, and trajectory sampling is performed within it. The number of trajectory lines for each training sample is randomly selected between 1 and 10.

### 4.2 Comparison

Among existing methods, Pika [1] is the most similar to ours. Pika can perform local video editing by defining an editing area. The difference is that Pika controls the new content in the editing area by text and has no motion control. In addition, the recent work AnyV2V [26] proposes editing the first frame of the video to achieve entire video editing, which has similarities with our ReVideo. InsV2V [10], using editing instructions to edit the video, can also maintain unedited content. Therefore, in this paper, we compare our ReVideo with these three methods. The visual comparison in Fig. 6 shows that in some fine-grained editing scenarios, such as putting sunglasses on a man, AnyV2V has a loss of edited content. In addition, the unedited area of InsV2V and AnyV2V suffers from content distortion. Although Pika can generate smooth and high-fidelity results, it is difficult to accurately customize new content by text, especially in adding new objects, *e.g.*, adding a dog on the soccer field. Adding new objects to the scene is also challenging for InsV2V. Due to the lack of motion

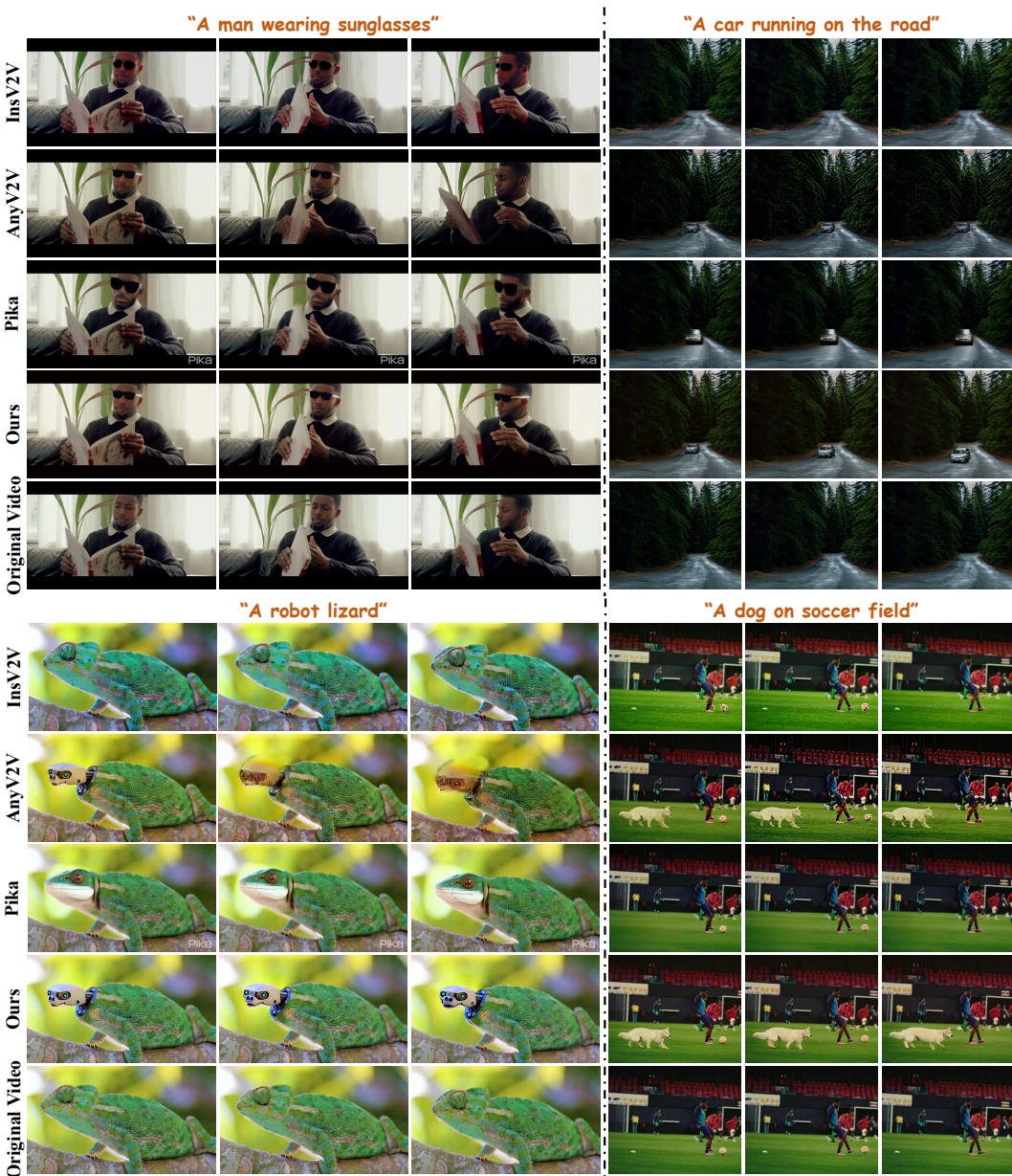

Figure 6: The visual comparison between InsV2V [10], AnyV2V [26], Pika [1], and our ReVideo.

control, AnyV2V and Pika usually produce static motion of the edited content, such as a car driving on the road. In comparison, our ReVideo can effectively broadcast the edited content throughout the entire video while allowing users to customize the motion in editing areas.

In addition to visual comparison, we employ automatic metrics and human evaluation to measure the performance of different methods. For this task, we build a test set containing 16 videos, with the resolution being $720 \times 1280$. Following previous works [7, 26], automatic metrics employ CLIP score [37] to measure text alignment and temporal consistency. The text alignment is obtained by calculating the average CLIP cosine similarity between each frame and editing description. Temporal consistency is computed by average CLIP cosine similarity between every pair of consecutive frames. We employ PSNR [43] to measure the reconstruction quality of unedited content. The human evaluation considers two aspects, *i.e.*, overall video quality, and whether the editing target is achieved. We allow 20 volunteers to choose the best method for each test sample on each aspect. The results in Tab. 1 show that our ReVideo performs better than InsV2V and AnyV2V in all evaluation terms.

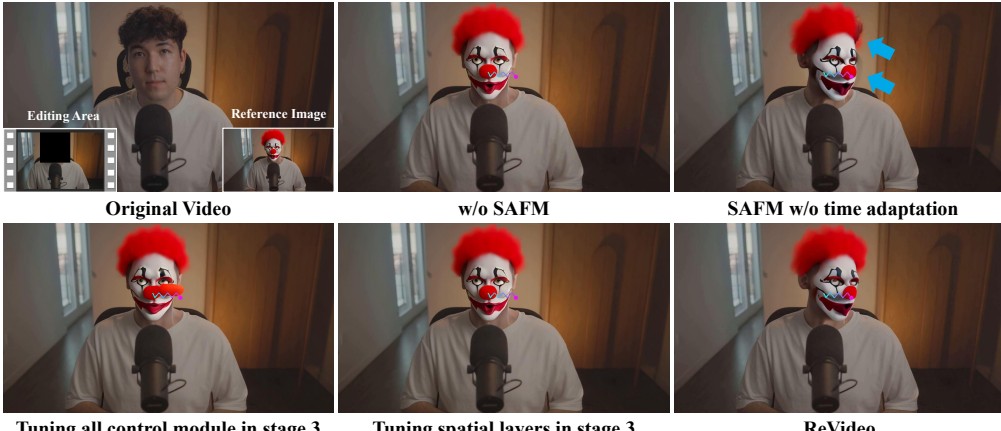

Figure 7: Ablation study of our ReVideo.

Compared with Pika, our performance is slightly lower in the evaluation of temporal consistency and the quality of unedited content. Notably, AnyV2V and Pika usually generate static motion of new content due to the lack of motion control. Static motion tends to score higher in consistency evaluation, measured by CLIP similarity of adjacent video frames. Our method has obvious advantages over Pika in text alignment and human evaluation, reflecting the significant gap between text-guided local editing and user-specified local editing. Our ReVideo can precisely specify the appearance and motion of the editing area, better meeting requirements for accurate customization.

The time complexity of different methods is also presented in Tab. 1. The experiment is conducted on an A100 GPU, with the video resolution being 768x768. Results show that our method has significantly lower time costs compared to other methods.

## 4.3 Ablation Study

In our ReVideo, we design the spatiotemporal adaptive fusion module (SAFM) to help decouple the control of unedited content and motion customization in diffusion generation. It predicts a fusion weight $\Gamma$ conditioned on the editing area $\mathbf{M}$ and time step t. Then, the fusion of content and motion control is achieved through Eq. 5. In this part, we conduct an ablation study on this fusion mechanism. In addition, we only fine-tune the key embedding and value embedding of the temporal self-attention layers in the SVD model and control module in the stage of deblocking training. In the ablation study, we discuss the impact of tuning parameters in deblocking training.

**The effectiveness of SAFM**. To demonstrate the effectiveness of SAFM, we replace SAFM with direct summing of motion and content control. The results in Fig. 7 show that the direct summing fusion cannot accurately control the motion in some complex motion trajectories, *e.g.*, wavy lines. In comparison, using SAFM can help decouple content and motion control in the editing area, achieving more accurate trajectory guidance.

**The effectiveness of time adaptation in SAFM**. We remove the time condition in the SAFM module, *i.e.*, using the same weight map $\Gamma$ to fuse content and motion control in each diffusion sampling step. The results in Fig. 7 show that not distinguishing $\Gamma$ in different sampling steps leads to unsatisfactory artifacts at the boundary of the editing area.

**Tuning parameters in deblocking training**. Although the training in stages 1 and 2 enables the control module to have good local motion control capabilities, we find that there is still an ignoring of motion control in the training of stage 3, *i.e.*, deblocking training. As shown in Fig. 7, the local motion control capability is degraded after we tune the entire control module in stage 3. Therefore, we optimize a part of the parameters to maintain the prior of local motion control. Experiments show that fine-tuning spatial layers still triggers the ignoring of motion control. In comparison, fine-tuning key embedding and value embedding of the temporal layer in the control module and the base model has minimal impact on local motion control capability. The edited and unedited areas are also harmoniously fused. More ablations of tuning parameters are presented in **Appendix**.

# 5 Conclusion

In this paper, we aim to solve the problem of local video editing. The editing target includes visual content and motion trajectory modifications. To the best of our knowledge, this is the first attempt at this task. In this new task, We find a coupling problem between unedited content and motion customization. Directly training these two control conditions on the video generation model will cause the ignoring of motion control. To address this issue, we develop a three-stage training strategy to combine these two conditions coarse to fine. In addition, we design a spatiotemporal adaptive fusion module to further decouple unedited content and motion-customized content in different diffusion sampling steps and spatial locations. Extensive experiments demonstrate that our ReVideo has promising performance on several accurate video editing applications, *i.e.*, (1) locally changing video content while keeping the motion constant, (2) keeping content unchanged and customizing new motion trajectories, (3) modifying both content and motion trajectories. Our method can also easily extend these applications to multi-area editing without specific training.

**Limitations**. Although our method can regenerate local areas of the video, the regeneration quality is limited by the base model. In some scenarios where the generation prior of SVD is not ideal, some unexpected artifacts may occur in the editing results.

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

# A Appendix

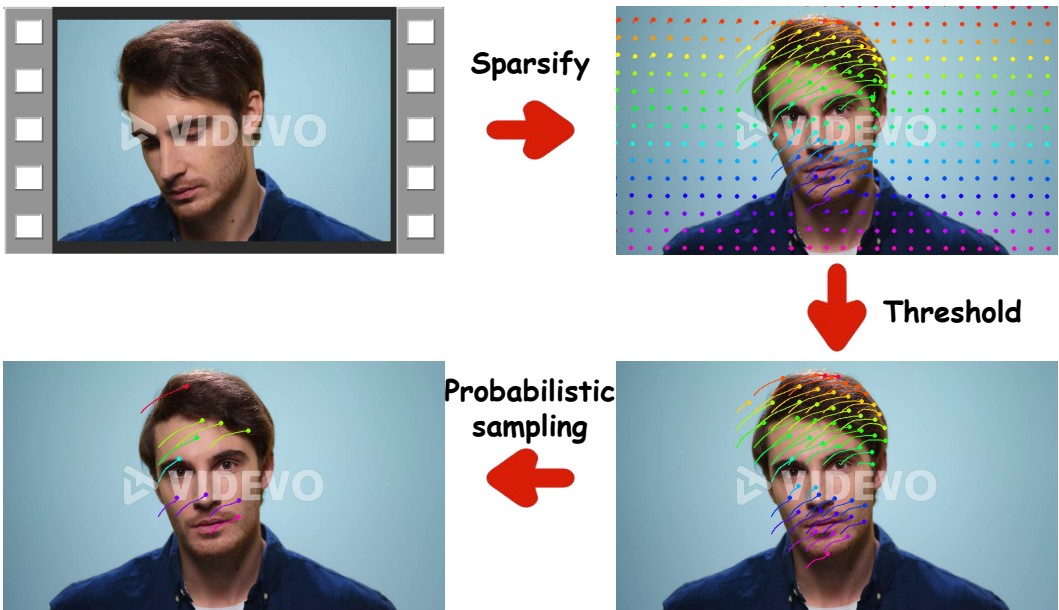

Figure 8: The trajectory sampling pipeline in ReVideo training.

## A.1 Details of Trajectory Sampling

As described in our main paper, trajectory sampling in the training process includes three stages, *i.e.*, sparsifying, threshold filtration, and probabilistic sampling. We present the visualization of this pipeline in Fig. 8. **In sparsifying**, we use a grid [57] to sparsify the dense sampling points, obtaining $N_{init}$ initial points. **In threshold filtration**, we use the mean of the tracking length of these $N_{init}$ points as the threshold $l_{Th}$ to filter out points with large motion. **In probabilistic sampling**, we use the normalized lengths of these sampling points as sampling probabilities to sample $N$ points from them. $N$ is randomly selected from 1 to 10.

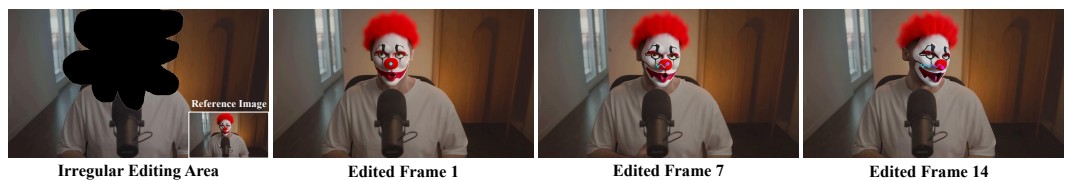

Figure 9: The robustness of our ReVideo for irregular editing areas.

## A.2 Robustness for Irregular Editing Area

In our main paper, we demonstrate the robustness of our method on multi-area editing without specific training. In Fig. 9, we present another robustness of our method for irregular editing areas. As can be seen, even though our method is trained on rectangular editing areas, it has stable content and motion editing capabilities when facing a hand-drawn irregular editing area.

## A.3 An Extension Application of Video Inpainting

In some video editing scenarios, specifying motion trajectories is challenging, such as when erasing an object in a video. We find that by leaving the motion control empty in these cases, our method can automatically generate the motion state of the editing area and match it with the motion in the unedited area. This is due to our inherent capability to predict the motion in the editing area via unedited content. An example is presented in Fig. 10.

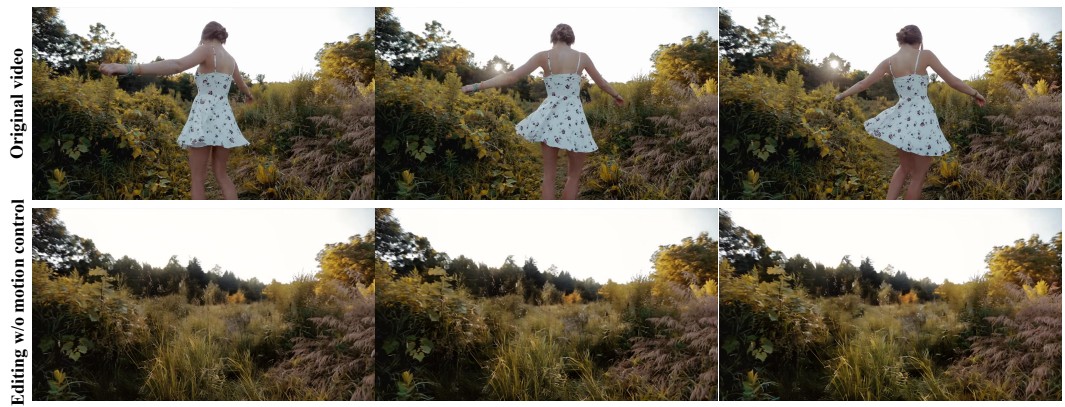

Figure 10: The application of object removing without specifying the motion trajectory. Note that the background is not static but is moving to the right.

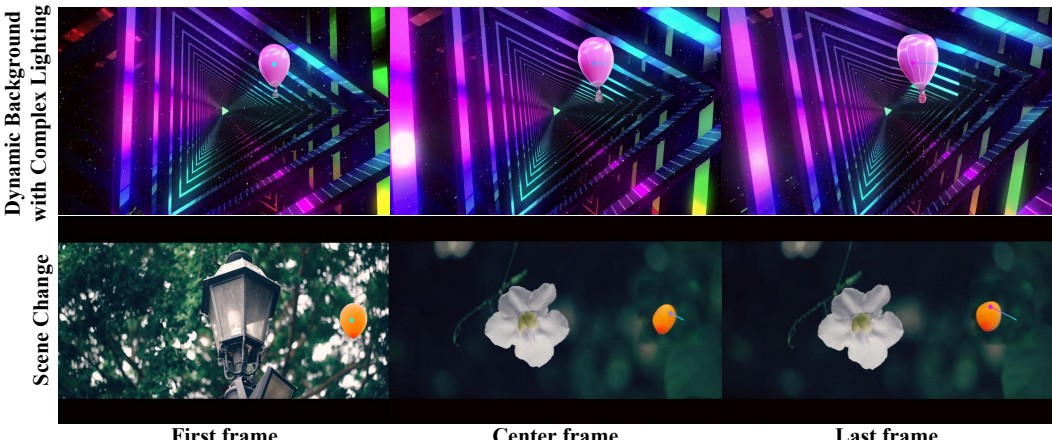

| First frame | Center frame | Last frame |

Figure 11: The editing results in some complex scenarios. The first row has dynamic background with complex lighting, and the second row has scene change.

## A.4 Additional Discussion in Complex Scenarios

In Fig. 11, we try some complex scenarios. The result shows that our method can handle dynamic lighting and texture, but scene change affects the content quality, which is a failure case.

## A.5 Details of Content Encoder and Motion Encoder

At the input of our spatiotemporal adaptive fusion module (SAFM), two encoders, *i.e.*, $E_c$ and $E_m$, separately encode the content and motion conditions. $E_c$ and $E_m$ have the same low-cost structure. This structure contains three sub-blocks, each consisting of a convolution and a downsampling operation, mapping the condition map to the same size as the latent $\mathbf{z}_t$.

## A.6 More Editing Results of ReVideo

In Fig. 12, we present more editing results of our ReVideo, including adding new objects to the video, modifying the motion trajectory of existing content in the video, editing existing content while maintaining the motion trajectory, and multi-region editing. As can be seen, the editing results achieve the editing goals, and motion control and content control coexist harmoniously.

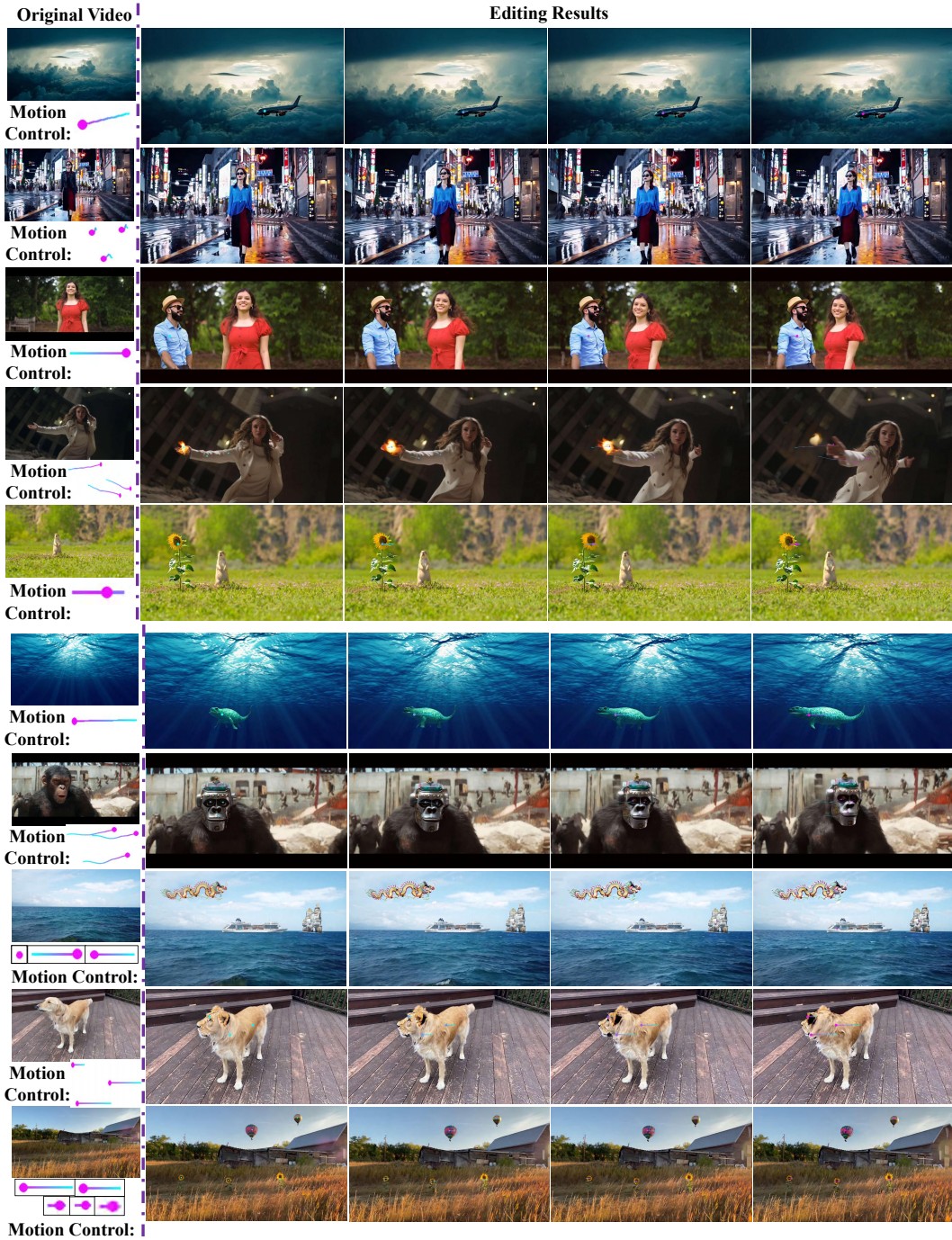

Figure 12: More editing results of our ReVideo.

## A.7 Necessity of Fine-tuning the Base Model

In the training process of stage 3, we fine-tune the key embedding $\mathbf{W}_k$ and value embedding $\mathbf{W}_v$ of the temporal self-attention layer in the control module and base model. In Fig. 13, we demonstrate the necessity of fine-tuning the base model in two scenarios. Specifically, in some complex scenarios, such as the forest shown in the first row, not fine-tuning the base model would result in content disjunction, *e.g.*, the misaligned tree trunk. The second row shows the case where there is a high coupling between the unedited content and editing content, such as retaining the motion of hair and only editing the facial movement. Fine-tuning the base model can alleviate artifacts brought by the

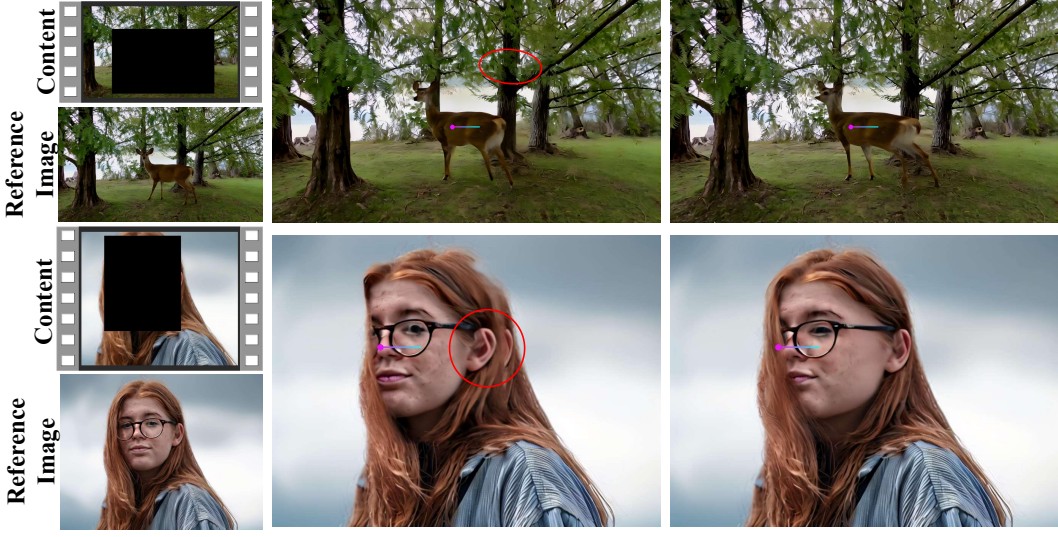

Figure 13: The necessity of fine-tuning key embedding and value embedding in the base model, *i.e.*, SVD.

motion conflict between the highly coupled edited and unedited areas. Therefore, jointly fine-tuning the base model helps to produce more harmonious editing results.

## A.8 More Frame Editing

In addition to editing a fixed number of frames based on the base model SVD, our ReVideo can process more frames. In implementation, we use the sliding window strategy, where the last frame of the editing result in the previous window is used as the reference image for the current window. Fig. 14 shows the editing results of our method on a 9-second video containing 90 frames. One can see that our ReVideo broadcasts the editing of the first frame into the 90-frame video while controlling the motion of the new content to be consistent with the original video. At the same time, we also observe that the error accumulation affects the editing quality. This is an inherent issue in long video editing, and a more powerful base model can alleviate this issue.

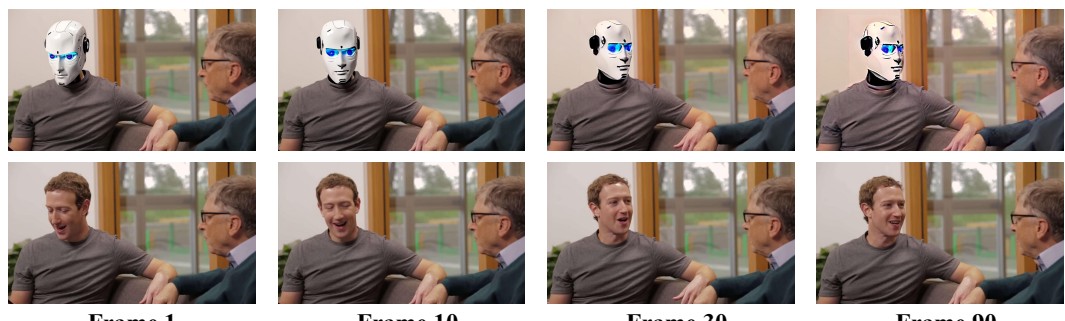

Figure 14: The ability of our ReVideo to extend the number of editing frames. The results demonstrate the performance of our ReVideo in processing a 9-second video containing 90 frames.

## A.9 More Discussion of Pika in Adding New Object

In the comparison section of the main paper, we find that Pika [1] has weak editing capabilities in adding new objects. To eliminate the influence of randomness, we generate 5 times with random seeds in the case of adding new objects in Fig. 15, *i.e.*, adding a plane in the sky. We set the strength of text

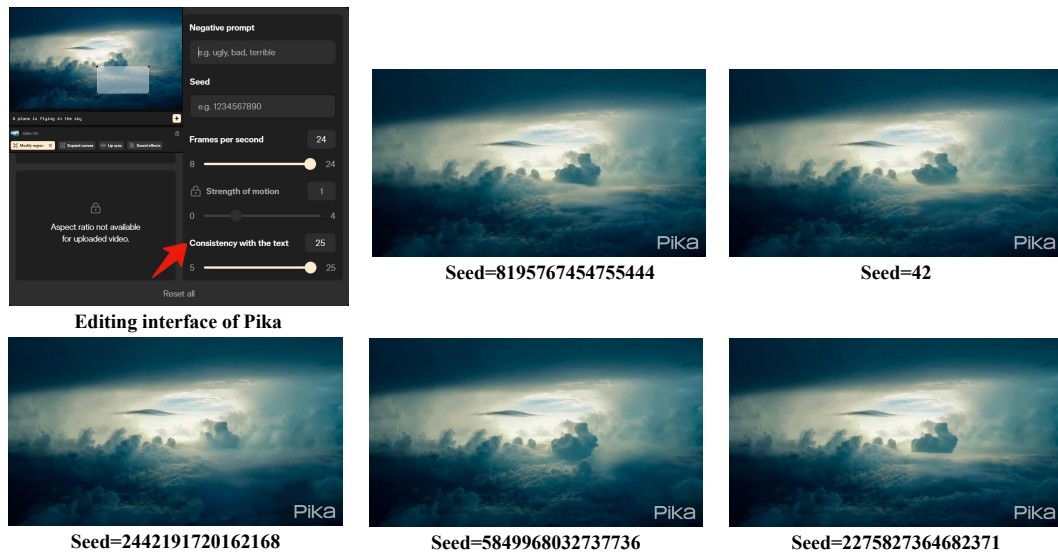

**Editing interface of Pika**    **Seed=8195767454755444**    **Seed=42**

**Seed=2442191720162168**    **Seed=5849968032737736**    **Seed=2275827364682371**

Figure 15: More failure cases of Pika in adding new objects to a video. We set the text consistency control parameter to the highest level during testing. The editing target is to add a plane in the sky.

consistency to the highest level, but all 5 editing results failed. This indicates the inaccuracy of text as the control signal of local redrawing. In comparison, editing the first frame and then broadcasting it to the entire video can accurately specify the content of the editing area.

