# OpenReview forum: "ReVideo: Remake a Video with Motion and Content Control"
_NeurIPS.cc/2024/Conference — NeurIPS 2024 poster_

### Official Review · Reviewer_YHcN · 2024-07-07

**Soundness:** 3
**Presentation:** 3
**Contribution:** 3
**Rating:** 7
**Confidence:** 5

**Summary:**

ReVideo presents a novel view of video editing by modifying content with input trajectory to create new content. It designs a three-stage strategy to wrestle out the problem of ignoring motion control when direct training. The main contribution of this work relies on the new task of editing motion via user-specified trajectory while keeping the original video movement. The editing results are superior and photorealistic.

**Strengths:**

1. The first video editing work on creating new motion and content.
2. Good writing; the paper is easy to follow, and the motivation and three-stage training strategy on decoupling content and motion control is reasonable. The proposed SAFM learned a dynamic fusion weight at different timesteps.
3. The editing results are photorealistic and adhere to the original motion or follow user-specified trajectory with no artifacts.

**Weaknesses:**

1. The author did not provide the method or explanation of how ReVideo edits the first frame, making the total editing pipeline not end-to-end for users.
2. Part of the original video motion, like mouth movement in the Zuckerberg->robot (head6) and tail movement in dog->lion, is not kept in the edited video.
3. I would like to know how the drag-based editing method handles non-rigid motion, such as the rotation of car tires from a side view. In examples like sea2 and sea2_2, where a shark and a dinosaur are added, the limbs of the animal seem unable to move, making the video look unrealistic. However, in soccer and some human-centric examples, the legs of dogs and people can move normally. Therefore, I would like the authors to add an example of a vehicle moving on the road from a side view, including the movement of the wheels, to address my concerns. This may be a limitation of the drag-based method.
3. There is no quantitative comparison of the ablation study; I understand that the image results in Fig 7 are clear, but only one video qualitative ablation is not reasonable.
4. There are no qualitative video comparisons with other methods in the supp or project page, but only Fig 6, and the automatic metrics are worse than pika even though I understand the clip scores are not accurate, which can not reflect temporal consistency accurately. I suggest the author supply the comparison video between Revideo and other methods in the rebuttal phase.
5. The training cost of three stages: even though Revideo makes great progress in creating new motion, training cost like GPU costs, time costs, memory costs and so on, is still a problem since users prefer to edit a video in a zero-shot manner when using a pretrained video generation model and the compared methods like AnyV2V is training-free.

**Questions:**

1. The method to edit the first frame needs to be declared.
2. Non-rigid motion-like side view wheels movement of cars.
3. Qualitative video comparisons with other methods.
4. The inference time/training cost comparison with other similar methods.
5. What about ReVideo performing in editing multiple objects simultaneously in the same video?
6. Can ReVideo work on text-to-video generation models?

**Limitations:**

no significant limitations.

---

> ### Author Rebuttal · Authors · 2024-08-06
>
> ## Q1 The Editing of the First Frame
> Thanks for this suggestion. The editing method for the first frame is arbitrary, like the setting in AnyV2V. The results presented in the paper utilize text-guided image inpainting tools and InstructPix2Pix. Note that in our framework, editing the first frame is not mandatory. Users can choose to keep the content and modify the motion of the video. We will add a detailed explanation of the editing workflow in the paper.
> ## Q2 Some Motion is Not Kept
> We need to clarify that our editing framework uses sparse motion control (trajectory lines) in the editing region, which is challenging to handle slight motions as you mentioned. To support such motion control, our framework may need to incorporate dense motion representations, such as optical flow. A potential solution is to mix sparse trajectories with dense optical flow during the training process, allowing our method to apply sparse and dense motion control selectively. We will explore this design in future work.
> ## Q3 Non-rigid Motion
> Thank you for this constructive suggestion. We add a video of editing a side-view driving car in **Fig.9** in the attached PDF. We can see that while the car's rigid body is moving, the car's wheels are also rolling (although it might not look entirely realistic). Similar phenomena can be observed in some demo videos in our supplementary, such as adding a tank on the desert. The tank's wheels are moving in a regular pattern, and the shadow behind the tank moves accordingly. We believe that this phenomenon, including the natural motion of dogs and humans that you mentioned, is due to the priors of physical world that the base model (SVD) learned from large-scale video data. The unnatural movement of the shark is also caused by the limitations of SVD's prior. To verify this, we use SVD to perform image-to-video generation based on the shark image. The result in **Fig.9** in the attached PDF shows that while the water surface has natural ripples, the shark remains stationary. We believe that by using a more powerful base model (such as the Sora architecture), we can achieve more natural editing results.
> ## Q4 Quantitative Comparison of the Ablation Study
> To address this concern, we measure the editing accuracy under different settings in our ablation study, as shown in the table below. We use a point tracking model (CoTracker) to extract trajectory in the edited results and then calculate the Euclidean distance with the target trajectory. We will add this quantitative comparison to our paper.
> |    | w/o SAFM | SAFM w/o time adaptation | Tuning all control module in stage3 | Tuning spatial layers in stage3 | ReVideo |
> |:-:|:-:|:-:|:-:|:-:|:-:|
> | Accuracy (Pixel)      |   37.34  |           8.92            |               42.26                |              44.78              |   5.21  |
> ## Q5 Qualitative Video Comparisons
> Yes, it is necessary to provide comparison videos of different methods. However, due to the rebuttal policy, we cannot provide external links. Instead, we select two videos to present in **Fig.10** in the attached PDF. One can see that AnyV2V has gradual changes in video content in some editing samplings, leading to instability. Pika and AnyV2V both tend to produce static editing results, due to the lack of motion control. While such static results may achieve higher scores in CLIP, it is not natural enough. Additionally, Pika tends to fail in some challenging editing scenarios, such as transforming a part of a lizard into a robot.
> ## Q6 Training Cost
> We agree that our implementation involves some training costs compared with training-free methods. However, our approach offers significant performance gains:
>
> (1) Our method allows precise customization of local content and motion in videos, which is not achievable with existing training-free methods.
>
> (2) Our method can produce high-quality and stable outputs. Training-free methods often struggle with video generation quality and stability of the edited content.
>
> (3) Our method has an advantage in inference complexity. Some training-free methods, such as AnyV2V, require a long time of ddim inversion to ensure the editing quality. The table below shows the time costs of different methods during the inference stage. The experiment is conducted on an A100 GPU, with the video resolution being 768x768. Results show that our method has significantly lower time costs compared to other methods. Therefore, we believe that our training cost (4 A100 GPUs) is reasonable and necessary. We will add the complexity analysis to our paper.
> |                | InsV2V | AnyV2V                     | ReVideo |
> |-|:-:|:-:|:-:|
> | Inference Time (s) |  132   | 303 (DDIM Inv) + 80.9 (Inference) |  26.8  |
> ## Q7 Performance in Editing Multiple Objects Simultaneously
> In the demo video in our supplementary, we show examples of editing six individual petals in the same video, as well as editing multiple flowers and balloons in one video. This demonstrates the robustness of our method in multi-object editing. Additionally, **R1** raises a concern in **Q.5** about the impact of the number of editing regions on performance. Results in **R1-Q.5** show that each editing region has a high independence, and the impact of the region numbers is almost negligible.
> ## Q8 Whether works on text-to-video models
> Yes, it is possible, but the implementation would be more complex than with an Image-to-Video model. The image condition port in the Image-to-Video model can naturally be used as the input of the edited first frame, whereas the Text-to-Video model lacks this port and would require additional training to incorporate it.

---

> ### Comment · Reviewer_YHcN · 2024-08-11
>
> My concerns have been thoroughly addressed in the rebuttal. Overall, this paper is well-structured, offering significant contributions and demonstrating impressive performance. It brings fresh perspectives to the research field, so I have decided to upgrade my final rating to accept.

---

> > ### Author Response · Authors · 2024-08-11
> >
> > Thank you for your efforts in the review, improving our paper to a higher standard. We will revise our paper based on your comments and suggestions.

---

### Official Review · Reviewer_ZSh9 · 2024-07-12

**Soundness:** 2
**Presentation:** 2
**Contribution:** 2
**Rating:** 5
**Confidence:** 3

**Summary:**

The paper presents a video editing method that enables precise localized adjustments to content and motion within specific areas of a video. It introduces a three-stage training strategy and a spatiotemporal adaptive fusion module to integrate edits across frames and locations effectively. This method allows for complex editing tasks such as changing content while maintaining motion, adding new motion to static content, and simultaneously modifying both elements.

**Strengths:**

- The paper introduces a novel challenge of editing both content and motion in specific video areas and combines techniques from diffusion models and video editing to achieve nuanced control.
- The three-stage training strategy enhances the robustness and effectiveness of the edits, supported by experimental validation that demonstrates superior performance compared to existing methods.
- The paper is well-organized and clearly explains complex concepts, including the innovative spatiotemporal adaptive fusion module and detailed training strategy.

**Weaknesses:**

- The decoupling training could cause some artifacts. Although the paper demonstrates these artifacts could mostly be alleviated by deblocking training. I can still see some blocky/unnatural results in the result videos.
- The training is quite complicated and separated into three stages. I feel the training strategy could 'overfit' this particular video dataset.
- This method is more like a direct combination of video diffusion and ControlNet.
- More detailed implementation specifics, particularly regarding parameter settings and the architecture of the spatiotemporal adaptive fusion module, are needed.
- The method's computational demands and potential scalability issues are not adequately addressed. For example, what kind of GPU does one need to perform training and testing?
- The paper focuses heavily on technical aspects with less consideration of user interaction.

**Questions:**

- What kind of GPU does one need to perform training and testing?
- Will the authors release the training and testing code along with pre-trained models upon acceptance?

**Limitations:**

Quality limited by SVD.

---

> ### Author Rebuttal · Authors · 2024-08-06
>
> ## Q1 About Artifact
> We agree that our method still has room for improvement. We want to clarify this concern from two points:
>
> (1) **The challenge of this task and our novelty.** Our method is the first attempt at local content and motion editing for videos. In Section 3.2 of the main paper, we conduct extensive toy experiments to demonstrate the challenge of this task, as it requires overcoming significant coupling between control conditions. In implementation, we meticulously design both the training strategy and the model architecture to address this challenging task.
>
> (2) **Editing Performance.** We present several useful applications and high-quality editing results in the paper, which is unachievable for existing methods with a significant gap. Although some artifacts exist in the edges, as you concerned, they are often difficult to detect in the result videos.
>
> We will continue to improve our method in the future.
>
> ## Q2 Overfitting in Training
> We need to clarify that although the three-stage training strategy might be a bit complicated, our extensive experiments in the paper (Section 3.2) demonstrate that the problem we face is indeed challenging. Therefore, we believe that using a progressive learning approach to tackle this difficult problem is reasonable, and the role of each training stage is demonstrated in the paper. We would greatly appreciate any references or suggestions you could provide to help us improve our current training strategy.
>
> In Section 4.1, we show that our training data includes 10 million videos from the WebVid dataset. Therefore, overfitting is not a concern under such a dataset scale. The test results also demonstrate the diversity of the editing scenarios.
>
> ## Q3 Like a ControlNet
> We need to clarify that ControlNet is a commonly used method for condition inputs. In this paper, we also introduce that our method uses conditional injection of ControlNet. Moreover, combining video diffusion with ControlNet cannot solve our problem, and it is not a contribution of this work. The key contribution of our work lies in the proposing of a novel training strategy to solve the unexplored fine-grained video editing task. Additionally, we design a spatiotemporal adaptive fusion module to improve the fusion of motion and content control.
>
> ## Q4 Detailed Implementation of SAFM
> Our SAFM consists of four convolutional layers, each followed by a SiLU activation layer, with a Sigmoid function applied at the end. The SAFM has 2 million parameters, which is negligible compared to the 671 million parameters in the SVD model. We will add this detailed description to the paper.
>
> ## Q5 Computation Demand
> In Section 4.1 of the paper, we describe that our training is conducted on four A100 GPUs. During the inference, it requires 22GB of VRAM to edit a 16x768x1344 video. In the table below, we present a comparison of computational complexity for different methods. The experiment is conducted on an A100 GPU, with the editing target being a 16x768x768 video. The results demonstrate the advantage of our method in inference efficiency. Therefore, we believe our approach is efficient and flexible. We will add a detailed description of complexity demand in the paper.
> |                | InsV2V | AnyV2V                     | ReVideo |
> |-|:-:|:-:|:-:|
> | Inference Time (s) |  132   | 303 (DDIM Inv) + 80.9 (Inference) |  26.8  |
>
> ## Q6 Focuses Heavily on Technical Aspects
> We want to clarify this concern via the interactive design and capability of our method. To offer an intuitive user interaction, we chose the sparse and easy-to-interact trajectory lines as the motion control. In the editing mode, users can selectively customize content or motion. In scenarios where motion control is difficult to specify, our method can also automatically generate the motion in the editing region. Therefore, our method is user-friendly in controlling inputs and functions. We will include a detailed description of user interaction in the paper.
>
> ## Q7 Open Source & Limitation
> Yes, we will open-source all the code.
>
> We need to clarify that our method is not limited to a specific base model. The reason we use SVD as the base model is that SVD is the best open-source model currently available to us. Using a better base model can further improve our editing results.

---

> ### Author Response · Authors · 2024-08-08
>
> Dear R2, we would like to know if your concerns are addressed. If any questions, we can discuss them in this phase.

---

> > ### Comment · Reviewer_ZSh9 · 2024-08-08
> >
> > I really appreciate the author's rebuttal. However:
> >
> > 1. The authors claim that the proposed method's novelty is beyond ControlNet. However, I am still not convinced. From Figure 2 in the paper, the two potential structures are just two slightly different ways of conditioning ControlNet. The trainable control module and zero convolution layers are exactly the same as ControlNet. The only difference is that you replace the condition (originally depth, canny edge map, normal, etc.) with a content map specifying the editing region and the motion trajectory. I still cannot say this analysis is a contribution. I'd say this is rather than an ablation on network architectural design.
> >
> > 2. In order to keep the original motion from the original video. This paper further proposes a data construction strategy, as shown in Figure 4, to decouple the motion between the editing region and the original region. However, this approach is still too naive, as it just performs CutMix [1] to combine two videos into one. However, in this case, the goal is to manipulate motion. I am still not convinced this stage is a contribution.
> >
> > 3. The training pipeline, separated into three stages, is very complex. There will be many hyperparameters that could influence the training in each stage. Also, the entire pipeline relies on many existing off-the-shelf techniques/datasets such as SVD and WebVid (This dataset is actually banned and no longer available due to some legal issue. Thus, the proposed method will not be reproducible. However, this is not the fault of this method.), and CoTracker. If more advanced techniques show up, one needs to tweak all the parameters of the entire training process to make the proposed method work. This makes this method not general
> >
> > 4. Following the previous comment, the proposed pipeline is more like an engineering work by combining existing techniques (SVD, ControlNet, CoTracker, CutMix) and using a massive computational power (4x A100), directly training for a long time (6 days) on a largescale dataset (WebVid). I really appreciate the edited video results. Some of them are amazing. In some of them, I still see artifacts (which might be the cons from SVD). In the entire paper, I do not see many novelties other than architectural designs and training data augmentation.
> >
> > Due to the above reasons, I am still leaning towards rejection. However, I would like to hear and discuss more with the authors and other reviewers.
> >
> > Best,
> >
> > Reviewer ZSh9
> >
> > [1] Yun, Sangdoo, et al. "Cutmix: Regularization strategy to train strong classifiers with localizable features." Proceedings of the IEEE/CVF international conference on computer vision. 2019.

---

> ### Author Response · Authors · 2024-08-08
>
> We need to clarify that:
>
> 1. **ControlNet is a commonly used method for condition injection, not a solution for our problem (as verified in Section 3.2). The contribution of this paper is to propose effective solutions for local video editing.** Additionally, we modify ControlNet by designing a region-adaptive fusion method to make the condition embedding more suitable for this task.
>
> 2. **Why is the decoupling training strategy not considered one of our contributions?** Dear reviewer, have you ever seen other similar solutions for video editing? The reference you provided is an image classification method—what relevance does it have to our paper? Just because two different images are mixed together?
>
> 3. **Our method is successfully validated on SVD and WebVid data, but this does not mean that our method can only be implemented using SVD and WebVid.** It can be applied to other data and base models. We believe that addressing a challenging problem by a progressive learning approach is reasonable. In many complex generation tasks, such as EMU2[1], the training difficulty is much greater than ours.
>
> [1] Generative Multimodal Models are In-Context Learners
>
> 4. Firstly, we are addressing an unexplored and challenging task that requires carefully designed training processes and fusion modules. This is not a simple engineering patchwork. Our core components did not previously exist. Secondly, what is the relevance of the CutMix to our work? Why can it replace our core contribution?
>
> Dear R2, we are eager to address these misunderstandings and look forward to your response.

---

> > ### Comment · Reviewer_ZSh9 · 2024-08-08
> >
> > 1. Novelty: Modifying existing methods doesn't automatically constitute novelty. Your paper shows good designs and results but appears incremental to ControlNet. Clarify what fundamentally differentiates your approach from ControlNet beyond architectural changes.
> > 2. Methodology: The cut-and-paste approach you're using is common in various video processing tasks, e.g., [1], not unique to your method. Referencing MixCut was to highlight this point, not to directly compare classification and video editing tasks.
> > 3. Generalizability: You claim your method can generalize beyond SVD and WebVid, but haven't demonstrated this. Without broader validation, the method's applicability to other datasets or base models remains unclear.
> > 4. Contributions: While some components in your work are new, they appear to be engineering efforts rather than fundamental contributions. The reference to CutMix was to point out similar data augmentation techniques in video processing, not to equate it directly with your method.
> >
> > [1] Liu, Yang, Zhen Zhu, and Xiang Bai. "Wdnet: Watermark-decomposition network for visible watermark removal." Proceedings of the IEEE/CVF winter conference on applications of computer vision. 2021.

---

> ### Author Response · Authors · 2024-08-08
>
> 1. ControlNet cannot accomplish our task. This is the fundamental difference. The following works all used ControlNet for conditional injection. We do not claim ControlNet as one of our contributions. It is a commonly used model, just like Stable Diffusion.
>
> [1] Control-A-Video: Controllable Text-to-Video Generation with Diffusion Models
>
> [2] ControlVideo: Training-free Controllable Text-to-Video Generation
>
> [3] CameraCtrl: Enabling Camera Control for Text-to-Video Generation
>
> [4] EVA: Zero-shot Accurate Attributes and Multi-Object Video Editing
>
> [5] CCEdit: Creative and Controllable Video Editing via Diffusion Models
>
> [6] Text-Animator: Controllable Visual Text Video Generation
>
> [7] IMAGE CONDUCTOR: PRECISION CONTROL FOR INTERACTIVE VIDEO SYNTHESIS
>
> 2. [1] is a paper on watermark removal, where a watermark image refers to the watermark being overlaid on the original image. This overlay is a task, not a solution. We still cannot find any connection between this and our method.
>
> [1] Liu, Yang, Zhen Zhu, and Xiang Bai. "Wdnet: Watermark-decomposition network for visible watermark removal." Proceedings of the IEEE/CVF winter conference on applications of computer vision. 2021.
>
> 3. Our method is the first attempt at this task and has already achieved state-of-the-art (SOTA) results. Even if adjustments are needed for other models and datasets, the issues found and solutions proposed in this work are still insightful.
>
> 4. CutMix you referenced is entirely different from our approach in both purpose and methodology. We do not think combining two images to enhance image classification performance can replace our contribution. What we propose is a motion-decoupled training strategy, not data augmentation. This decoupling strategy is effective and does not exist in our community.

---

> > ### Comment · Reviewer_ZSh9 · 2024-08-11
> >
> > I found out I severely misunderstood the operation of the motion decoupling part. The goal here is to manipulate different motions between the foreground (editing region) and background. Then, the purpose is largely different from MixCut or other cut-and-paste approaches. Could the authors further claim the purpose of the motion decoupling? Is my current understanding correct?

---

> > > ### Author Response · Authors · 2024-08-11
> > >
> > > This understanding is correct. In our experiments, we find a significant motion coupling between the foreground (editing region) and the background (please refer to Section 3.2). The model tends to learn how to estimate the motion state of the foreground (editing region) by relying on the motion in the background, thereby neglecting the condition of motion trajectory. To address this issue, we propose the motion decoupling training as you mentioned. We may not describe this clearly enough in the paper, and we will make a revision for this part.

---

> ### Comment · Reviewer_ZSh9 · 2024-08-11
>
> Dear authors,
>
> Thank you very much for the clarification. However, I still feel the paper is somewhat incremental by combining a ControlNet-like motion injection module with augmented (motion) training data. The approach and task themselves are new and novel. Furthermore, the result videos are impressive, with barely noticeable artifacts. I would like to raise my rating to borderline accept.
> Thanks again to the authors for the clarification and discussions!

---

> > ### Author Response · Authors · 2024-08-11
> >
> > Thank you for your efforts in the review. Your comments help improve our method, and we will continue to enhance our approach in future work.

---

### Official Review · Reviewer_F4MH · 2024-07-13

**Soundness:** 3
**Presentation:** 3
**Contribution:** 3
**Rating:** 6
**Confidence:** 3

**Summary:**

This paper presents ReVideo, a new approach for precise local video editing of both content and motion. It introduces a coarse-to-fine training strategy to progressively decouple content and motion control, and a spatiotemporal adaptive fusion module to integrate them effectively. Experiments show ReVideo can modify local video content, customize motion trajectories, or change both simultaneously, and extend to multi-region editing.

**Strengths:**

- This appears to be the first attempt at exploring local editing of both content and motion in videos using diffusion models. Being able to modify content and motion trajectories in specific regions is a novel capability compared to prior work.
- The proposed three-stage coarse-to-fine training strategy to progressively decouple content and motion control is an interesting technical approach to deal with the core challenge.
- The spatiotemporal adaptive fusion module is another novel component to integrate the content and motion conditions across sampling - steps and spatial locations.
- Extending the approach to allow multi-area editing without requiring specific training demonstrates flexibility.
- Most of the visual and quantitative results show improvements over prior methods

Overall, this paper addresses a timely and important topic with significant potential benefits for the community. Despite some weaknesses, the reviewer recommends acceptance, considering this is a relatively new area and the paper presents promising results. The score may be adjusted based on the quality of the rebuttal.

**Weaknesses:**

## Practicality of the Editing Workflow

The current editing interface requires users to specify both a target content image and a set of motion trajectories. While this allows for fine-grained control, it may not be the most intuitive or efficient workflow for common editing tasks. Consider the scenario of object removal - the user would need to carefully craft a content image with the object removed and ensure that the remaining motion trajectories are consistent. An alternative approach could be to directly specify the regions to remove and have the model infer the appropriate content and motion changes automatically. The paper would benefit from a more detailed discussion of the practical trade-offs and usability considerations of the proposed editing framework.

## Limited Motion Control
While the method allows for editing the motion of individual objects, it assumes that the overall scene motion (camera movement, background motion) remains fixed. This limits the applicability of the approach in scenarios where the goal is to modify the global motion patterns (e.g. stabilizing shaky footage, changing the camera viewpoint).

## Precise Placement and Key Contributions of this Paper

While the individual technical components (e.g. coarse-to-fine training, adaptive fusion) are well-motivated, it's worth considering whether similar strategies have been explored in related domains. For instance, progressive training to handle multi-factor variation has been used in GANs, and spatially-adaptive normalization is common in style transfer. Drawing more connections to such related work would clarify the novelty of the specific adaptations made here.

## Content-Motion Entanglement

- The key technical contribution of the paper is the decoupling of content and motion information through a coarse-to-fine training strategy. However, it's not clear if this decoupling is complete or if there are still some residual entanglements between the two factors. For instance, the edited content may still contain some motion information that could interfere with the specified motion trajectories, leading to artifacts or inconsistencies. A more thorough analysis of the content-motion separation and its impact on the editing quality would be informative.

- Is decoupling content and motion the only way to address the issue - could a joint representation learning approach work instead? Acknowledging alternate strategies would help justify the chosen approach.

- **Figure 4 is not very intuitive. It would benefit from additional justification, theoretical analysis, and insights into why such a simple composition from two videos is effective.** This is a key concern.

## Multi-area Editing
- The extension to multi-area editing is a nice addition, but the paper could go further in characterizing the challenges involved. Are there issues with preserving global coherence across multiple edited regions? How does the method scale with the number of regions? Providing such details would give a more complete picture of the capability.

## Clarity and Reproducibility
- Implementation details: There are some missing specifics that could hamper reproducibility. For instance:

   - How exactly are the editing regions defined during training - what is the procedure for randomly sampling them?
   - What metrics are used for the "threshold filtering" of motion trajectories and how were the thresholds chosen?
   - Are there any data augmentation, regularization or optimization tricks used during training?

## Evaluation Metrics

The quantitative evaluation relies primarily on low-level metrics like PSNR and LPIPS, which may not fully capture the perceptual quality and coherence of the edited videos. Additional metrics could provide a more comprehensive assessment:

- Metrics that specifically measure the consistency of the edited regions with the target content and motion (e.g. using an object detector or tracker).
- Metrics that evaluate the temporal stability and smoothness of the edited videos (e.g. some metrics that are used in video inpainting tasks, Please refer to [this repo](https://github.com/MichiganCOG/video-inpainting-evaluation) for details).
- Human evaluations of the overall realism, coherence, and faithfulness to the editing inputs (e.g. through user studies).



## Robustness Evaluation and Ablation Studies

While the paper does include ablations for a few key components (e.g. SAFM, training stages), there are other design choices that are not fully explored. For instance:

   - How important is the choice of motion representation (trajectory vs. alternatives)? Testing with different motion inputs would reveal the sensitivity to this factor.
   - What is the impact of the trajectory sampling strategy and hyperparameters? Varying the number and selection of trajectories could provide insight into the robustness.
   - How does the performance vary with the size and shape of the editing regions? A systematic evaluation across different region properties would be informative.
   - Only the end-to-end video editing pipelines are compared, but not the individual technical components. For instance, how does SAFM compare to simpler fusion schemes used in prior work?
  - Input noise and perturbations (e.g. in the content image or motion trajectories)

## Dataset Complexity

-  While the approach achieves good results on the chosen datasets, it's unclear how well it would generalize to more complex video content (e.g. with dynamic backgrounds, scene changes, occlusions etc.). Discussing the potential failure modes and current limitations would help scope the contribution appropriately.

- The examples shown in the paper are largely limited to simple object-level edits in relatively constrained scenarios (e.g. clean backgrounds, single objects). It's unclear how well the method would perform on more challenging videos with complex scenes, multiple objects, occlusions, camera motion, etc. Testing on a wider range of video complexity would help establish the generality of the approach.

## Editing Scenarios
The paper demonstrates a few key editing applications (e.g. object addition/removal, motion editing), but there are other important scenarios that are not explored, such as: performing semantic-level edits (e.g. changing the action or interaction between objects).
Showcasing the method's performance across a fuller range of editing tasks would demonstrate its versatility.

## Open Source
Will the code for training and inference be released?

**Questions:**

Please refer to the weakness section.

**Limitations:**

Please refer to the weakness section.

---

> ### Author Rebuttal · Authors · 2024-08-06
>
> ## Q1 Practicality of Workflow
> **Fig.1** and **Fig.2** in the attached PDF show that our method can still produce smooth results without specifying trajectory. This is due to our inherent capability to predict the motion in editing area via unedited content, enabling automatic motion generation when trajectory is difficult to specify. However, the content reference is essential in current framework. We will discuss the practicality in our paper.
> ## Q2 Limited Motion Control
> We need to clarify that global motion pattern in the editing region is not our editing target but needs to be consistent with unedited areas. The consistency is automatically produced without fixed motion assumption.
> ## Q3 Precise Placement
> Thanks for the suggestion. We will discuss related works in progressive learning and multi-condition control.
> ## Q4 Entanglement
> - Residual entanglement. Yes, it still exists and can be observed by reducing the weight of motion condition (**Fig.3** in the PDF). We believe it is useful and mild. **(1)Useful.** Inspired by **Q.1**, the entanglement can automatically produce the motion, when it is difficult to specify the motion of editing region. **(2)Mild.** Various editing results show that this entanglement is almost unnoticeable in default inference.
> - Other solutions. Yes, that's certain. But we think decoupled representations are necessary.
> - Decoupling training. The insight of why it works mainly comes from Sec.3.2 in our paper. In Sec.3.2, we use several toy experiments to show the strong coupling issue. The method in Fig.4 ensures no motion correlation between the editing and unediting region, preventing the model from estimating the motion in editing region via unedited content. We will add a more intuitive explanation in the paper.
> ## Q5 Multi-area Editing
> In **Fig.4** in the PDF, we select a complex wavy line as the editing target and gradually add editing regions. We compute editing accuracy by Euclidean distance between the trajectory in editing result (extracted by CoTracker) and the target. Results below show slight impact of region numbers on accuracy and consistency. We will add this in our paper.
> |Region Number|1|2|4|8|10|
> |-|:-|:-|:-|:-|:-|
> |Acc.|4.68|4.57|4.67|5.15|5.40|
> |Temporal Consistency|0.9926|0.9920|0.9918|0.9905|0.9908|
> ## Q6 Clarity
> In Sec.3.2 and A.1 of our paper, we describe the trajectory sampling. Threshold $l_{Th}$ is empirically defined as the mean length of sparsed trajectories. Each video has one editing area, the minimum bounding rectangle of sampled trajectories. It is expanded to at least 64x64 pixels. Data augmentation includes sampling frames at varying intervals and randomly reversing the video. We will update these details in our paper.
> ## Q7 Metrics
> - Measurement in editing region. Using a tracker helps evaluate our motion control, especially in ablation. Other methods do not have motion control. The content consistency in editing region can be compared among different methods. Below table shows CLIP Score between the editing region and target description.
> |Methods|InsV2V|AnyV2V|Pika|ReVideo|
> |-|:-:|:-:|:-:|:-:|
> |Local CLIP Score|0.2255|0.2378|0.2402| 0.2516|
> - Temporal stability. In our paper, we use CLIP-Image score to evaluate the temporal consistency, which is used in several related works (e.g., AnyV2V, Pix2Video). The reference repo is helpful, but most metrics need a target video, which is unachievable in editing task. After reviewing, PSNR Consistency can measure the temporal stability, as shown below.
> |Methods|InsV2V|AnyV2V|Pika|ReVideo|
> |-|:-:|:-:|:-:|:-:|
> |PSNR Consistency|34.94|34.12|40.51|39.26|
> - Human evaluation. In Tab.1 in our paper, we conduct a user study for overall video quality and editing accuracy. We agree that more evaluation terms is helpful. We will extend it accordingly.
> ## Q8 Robustness and Ablation
> - Choice of motion representation. Yes, trying more motion representations is helpful. But we only find the trajectory that meets our sparse and interactable requirement.
> - Impact of trajectory sampling. Training with significantly moved trajectory is crucial. Below table shows its significant impact. In contrast, trajectory numbers have little impact.
> ||Random sampling|ReVideo|
> |-|:-:|:-:|
> |Acc.|13.38|5.21|
> - Impact of size and shape. In Sec.A.2 of our paper, we show our support for irregularly shaped regions. Below table shows the performance after randomly expanding the width/height of editing regions on test set. The size and shape have little impact on performance, even when the editing region is entire video.
> ||Default|+$l\in$[32,64]|+$l\in$[64,128]|Global|
> |-|:-:|:-:|:-:|:-:|
> |Acc.|5.21|5.06|5.32|5.48|
> - Ablation of SAFM. "w/o SAFM" in Sec.4.3 is a simpler fusion scheme that fuses motion and content embedding by adding and convolution. It fails in complex motion control. This result and weight visualization in Fig.5 can verify the need of region-adaptive fusion.
> - Input perturbations. We try two content perturbations in **Fig.5** in the PDF. The base model's prior on high-quality videos can mitigate content degradation, and motion control works normally. If the trajectory is perturbed, the motion will follow the perturbation.
>
> We will update the above discussion in our paper.
> ## Q9 Dataset Complexity
> In **Fig.6** and **Fig.7** in the PDF, we try some complex scenarios. Our method can handle dynamic lighting and texture, but scene change affects the content quality, which is a failure case. We will include a discussion in our paper.
> ## Q10 Editing Scenario
> Our test has simple semantic interactions, such as wearing a hat or glasses. In **Fig.8** in the PDF, we test a more complex interaction where two balloons collide. They do not bounce off each other but overlap. This failure is because our base model (SVD) is weak in physical world modeling. We believe our method can achieve such interactions with a more powerful base model. We will discuss this failure in the paper.
> ## Q11
> Yes, we will release all code.

---

> > ### Comment · Reviewer_F4MH · 2024-08-09
> >
> > Thank you for your rebuttal and efforts to address the concerns raised. I still believe this work is timely and addresses an important topic.
> >
> > After considering the comments from other reviewers and your rebuttal, my opinion and suggestions are as follows:
> >
> > 1. The task scope and editing capabilities need to be defined more precisely. The initial submission did not clearly discuss the limitations and capabilities in the introduction and conclusion sections.
> >
> > 2. While the decoupling method is well-motivated, the proposed solution is straightforward and similar to proposals seen in many common computer vision tasks. There is little technical analysis and theoretical explanation provided, especially regarding potential artifacts introduced by the proposed method.
> >
> > 3. Given the current state of the field, the architecture is reasonable and enables motion control by trajectory, despite its complexity and the need for tuning many hyperparameters. However, a more concise solution may emerge in the future. To facilitate reproducibility, I recommend providing more detailed documentation of the training details in the paper.

---

> > > ### Author Response · Authors · 2024-08-09
> > >
> > > Thank you for your constructive suggestions. Your very detailed comments in the review improve our paper to a higher standard. We will make a revision accordingly.

---

### Author Rebuttal · Authors · 2024-08-06

We appreciate the efforts of all the reviewers, ACs, and PCs. We have carefully read and addressed all concerns. **Since we are limited to 6,000 characters per reviewer during the rebuttal phase, we could only provide brief responses to some questions.** If there are any further issues, we are happy to address them during the discussion phase.

We include the figures and videos in the attached PDF. Please note that you need to use **Adobe Acrobat** to open the PDF to watch the videos.

---

### Author Response · Authors · 2024-08-09

Dear reviewers, I would like to further explain our proposed three-stage training strategy. Its contributions mainly lie in two aspects:

1. We discover a significant coupling between content and motion in this task, which is an attribute difficult to detect. As described in Section 3.2, we find this attribute/issue after many failed attempts. This priori exploration can inspire future research in this community.

2. The proposed method successfully tackles an unexplored and challenging task. Each stage is validated and analyzed in Section 3.2. The design of each stage is effective and innovative given the current state of this research field.

---

### Decision · Program_Chairs · 2024-09-25

**Decision:**

Accept (poster)

**Comment:**

Overall, the reviewers agree that the paper tackles an unexplored and challenging task, significantly advancing the capability to modify content and motion trajectories in specific regions using diffusion models. The experiments demonstrate promising results and show improvements over prior methods. The reviewers have offered suggestions for revision, including a more precise description of the task scope and editing capabilities, a discussion of potential artifacts introduced by the proposed method, and detailed documentation of the training process. Additionally, to facilitate reproducibility and further research on this new task, it is strongly recommended that the authors release the code for training and inference, as they have committed to doing.